

**Landfast ice thickness in the Canadian Arctic Archipelago from Observations and Models**
Stephen. E. L. Howell[1], Frédéric Laliberté[1], Ron Kwok[2], Chris Derksen[1] and Joshua King[1]
[1]Climate Research Division, Environment Canada, Toronto, Canada
[2]Jet Propulsion Laboratory, California Institute of Technology, Pasadena, California, USA
**Abstract**
Observed and modelled landfast ice thickness variability and trends spanning more than five
decades within the Canadian Arctic Archipelago (CAA) are summarized. The observed sites
(Cambridge Bay, Resolute, Eureka and Alert) represent some of the Arctic's longest records of
landfast ice thickness. Observed end-of-winter (maximum) trends of landfast ice thickness (1957-
2014) were statistically significant at Cambridge Bay (-4.31±1.4 cm decade$^{-1}$), Eureka (-4.65±1.7
cm decade$^{-1}$) and Alert (-4.44±1.6 cm decade$^{-1}$) but not at Resolute. Over the 50+ year record, the
ice thinned by ~0.24-0.26 m at Cambridge Bay, Eureka and Alert with essentially negligible
change at Resolute. Although statistically significant warming in spring and fall was present at all
sites, only low correlations between temperature and maximum ice thickness were present; snow
depth was found to be more strongly associated with the negative ice thickness trends. Comparison
with multi-model simulations from Coupled Model Intercomparison project phase 5 (CMIP5),
Ocean Reanalysis Intercomparison (ORA-IP) and Pan-Arctic Ice-Ocean Modeling and
Assimilation System (PIOMAS) show that although a subset of current generation models have a
'reasonable' climatological representation of landfast ice thickness and distribution within the
CAA, trends are unrealistic and far exceed observations by up to two magnitudes. ORA-IP models
were found to have positive correlations between temperature and ice thickness over the CAA, a
feature that is inconsistent with both observations and coupled models from CMIP5.



## 1. Introduction

Landfast sea ice is immobile ice that is grounded or anchored to the coast [*Barry et al.,* 1979]. In the Arctic, this ice typically extends to the 20-30 m isobath. It melts each summer and reforms in the fall but there are regions along the northern coast of the Canadian Arctic Archipelago (CAA) where multi-year landfast ice (also termed an "ice plug") is present. The two most prominent regions of multi-year landfast sea ice in the CAA are located in Nansen Sound and Sverdrup Channel [*Serson*, 1972; *Serson*, 1974] (Figure 1). It has been documented that ice remained intact from 1963-1998 in Nansen Sound and from 1978-1998 in Sverdrup Channel [*Jeffers et al.,* 2001; *Melling*, 2002; *Alt et al.,* 2006]. The extreme warm year of 1998 disintegrated the ice in both regions and their survival during the summer melt season in recent years has occurred less frequently [*Alt et al.*, 2006]. Over the entire Arctic, landfast ice extent is declining at 7% decade$^{-1}$ since the mid-1970s [*Yu et al.,* 2013]

Records of landfast ice thickness provide annual measures of ice growth that can also almost entirely be attributed to atmospheric forcing with negligible deep ocean influence on local ice formation. While the key forcings on landfast ice and offshore ice are different, the seasonal behavior of landfast ice can nevertheless provide useful information for understanding the interannual variability of ice thickness in both regimes. Presently, there is no pan-Arctic network for monitoring changes in landfast ice but available measurements suggest thinning in recent years. Thickness measurements near Hopen, Svalbard revealed thinning of landfast ice in the Barents Sea region by 11 cm decade$^{-1}$ between 1966 and 2007 [*Gerland et al.,* 2008]. From a composite time series of landfast ice thickness from 15 stations along the Siberian coast, *Polyakov et al.* [2010] estimate an average rate of thinning of 3.3 cm decade$^{-1}$ between the mid-1960s and early



2000s. Relatively recent observations by *Mahoney et al.* [2007] and *Druckenmiller et al.* [2009]
found longer ice-free seasons and thinner landfast ice compared to earlier records.
At four sites in the CAA, *Brown and Cote* [1992] (hereinafter, BC92) provided the first
examination of the interannual variability of end-of-winter (maximum) landfast ice thickness and
associated snow depth over the period 1957-1989. Their results highlighted the insulating role of
snow cover in explaining 30-60% of the variance in maximum ice thickness. Similar results were
also reported by *Flato and Brown* [1996] and *Gough et al.* [2004]. In the record examined by
BC92, no evidence for systematic thinning of landfast ice in the CAA was found. Landfast ice
thickness records at several of these CAA sites are now over 50 years in length, which represents
an addition of more than two decades of measurements since BC92 during a period that saw
dramatic reductions in the extent and thickness of Arctic sea ice [e.g. *Kwok and Rothrock*, 2009;
*Stroeve et al.,* 2012].
The sparse network of long term observations of snow and ice thickness in the Arctic
(clearly exhibited by only four ongoing measurements sites operated by Environment Canada in
the CAA) has made the use of models imperative to provide a broader regional scale perspective
of sea ice trends in a warming climate. Given the coarse spatial resolution of global climate models,
previous studies focusing on the CAA have relied on either a one-dimensional thermodynamic
dynamic model [*Flato and Brown*, 1996; *Dumas et al.,* 2006] or a regional three-dimensional ice-
ocean coupled model [e.g. *Sou and Flato*, 2009]. Specifically, *Dumas et al.* [2006] found projected
maximum ice thickness decreases of 30 cm by 2041-2060 and 50 cm by 2081-2100 and *Flato and*
*Sou* [2009] reported a potential 17% decrease in overall ice thickness throughout the CAA by
2041-2060. However, in recent years some global climate models, reanalysis products, and data



assimilation systems are now of sufficient spatial resolution to assess potential landfast ice
thickness changes within the CAA.

This analysis examines the trends of measured landfast ice thickness, snow depth and air

temperature over a 50+ year period between 1957 and 2014 and compares the results with the
earlier analysis by BC92. We then use this observational foundation to evaluate the
representativeness of landfast ice in state-of-the-art global climate models, assimilation systems
and re-analysis products.

**2. Data Description**
**2.1. Observations**

Landfast ice thickness and corresponding snow depth measurement have been made

regularly at many coastal stations throughout Canada since about 1950. These data are quality
controlled and archived at the Canadian Ice Service (CIS) and represent one of the few available
sources of continuous ice thickness measurements in the Arctic. In general, thickness
measurements are taken once per week, starting after freeze-up when the ice is safe to walk on and
continuing until breakup or when the ice becomes unsafe. Complete details of this dataset are
provided by Brown and Cote (1992) and the dataset is available on the CIS web site
(http://www.ec.gc.ca/glaces-ice/, see Archive followed by Ice Thickness Data). Four sites in the
CAA were selected for study: Alert, Eureka, Resolute, and Cambridge Bay (Figure 1). Although
there are other sites in the database, these sites are the only ones than span the same 55-year period
between 1960 and 2014. The record at Mould Bay, used in BC92, terminated in the early 1990s.
Together these sites cover ~20° in latitude (Figure 1) that are adjacent to an area of thick Arctic
sea ice that experienced the highest thinning in recent years [*Kwok and Rothrock*, 2009; *Laxon et*



*al.,* 2013]. Values of maximum or end-of-winter ice thickness and corresponding snow depth
during the ice growth season were extracted from the weekly ice and snow thickness data at the
selected sites. As this study is concerned with annual variability in maximum ice thickness, the
main period of interest extends from September to late May.
The other source of observed data used in this study were monthly mean air temperature
records at Alert, Eureka, Resolute, and Cambridge Bay for which a complete description is
provided by *Vincent et al.* [2012].

**2.2. Models**
The representation of CAA landfast sea ice thickness within the Coupled Model
Intercomparison project phase 5 (CMIP5) is analyzed using the 1980-2005 Historical experiment
followed by the 2006-2099 Representative Concentration Pathway 8.5 (RCP85) experiment
[*Taylor et al.,* 2012] (Table 1). Monthly sea ice thickness (variable *sit*), sea ice concentration
(variable *sic*), 2 meter temperature (variable *tas*) and snow depth (variable *snd*) were used. The
CMIP5 data were retrieved from the British Atmospheric Data Centre database and accessed
through the Center for Environmental Data Analysis (www.ceda.ac.uk). Ensemble r6i1p1 and
r7i1p1 from model EC-EARTH were removed because of corrupted data. We obtain the multi-
model mean of trends at each grid point by creating the distribution of trends through a Monte-
Carlo simulation. We use a t-distribution for the interannual variability and build a noise model to
account for internal variability as in *Swart et al.* [2014] and *Laliberté et al.* [2016]. The multi-
model mean and its statistical significance is then obtained from the distribution. We obtain the
multi-model mean of Pearson correlations by first performing a Fisher transform and then apply





the same method as for the trends. The inverse Fisher transform is applied after obtaining the multi-
model mean and its significance.

We also investigate ice thickness values from a selection of the highest resolution models

[*Storto et al.,* 2011; *Forget et al.,* 2015; *Haines et al.,* 2014, *Zuo et al.,* 2015; *Masina et al.,* 2015]
from the Ocean Reanalysis Intercomparison (ORA-IP) [*Balsameda et al.,* 2015; *Chevallier et al.,*
2016] (Table 2) and from the Pan-Arctic Ice-Ocean Modeling and Assimilation System (PIOMAS)
[*Zhang and Rothrock*, 2003]. Supporting 2 meter temperature data was obtained from ERA-
Interim [*Dee et al.,* 2011].

**3. Results and Discussion: Observations**
**3.1. Climatology**

The average behavior of landfast ice at the four sites over the 50+ year record is

summarized in Table 3. Ice growth, approximately linear through most of the season, slows after
March (Figure 2). Ice thickness reaches a maximum of ~2-2.3 m by late May at all sites. Values
are consistent with that reported by BC92 and with recent observations of *Melling et al.* [2015]
and *Haas and Howell* [2015]. The standard deviations are nearly uniform (at ~0.2 m) across all
sites, giving a relatively low coefficient of variation (COV; a measure of relative dispersion
defined as the ratio of the standard deviation to the mean) of ~0.1. The thickest ice is found in
Eureka with a 1957-2014 mean of 2.27 m that is likely due to climatologically lower air
temperatures in the fall and winter (Table 3).

Snow depth also appears to grow linearly through the season, peaking in May but unlike

ice thickness the monthly variability is high (COV ~0.4) (Figure 3).  Mean October to May snow
depths at Resolute, Eureka and Alert range from ~18-23 cm compared to only ~8 cm at Cambridge



Bay (Table 3). The rapid buildup of the snow cover due to storms in the fall and early winter that
is evident over the Arctic Ocean multi-year ice cover [*Warren et al.,* 1999; *Webster et al.,* 2014],
is not seen in these snow depth records within the CAA. The linear behavior in snow depth is likely
maintained by continuous wind-driven redistribution and densification throughout the ice growth
season [BC92; *Woo and Heron*, 1989].

**3.2. Trends**

The time series of maximum ice thickness at Cambridge Bay, Resolute, Eureka and Alert

are illustrated in Figure 4 and summarized in Table 1. Statistically significant (95% or greater
confidence level) negative maximum ice thickness trends are present at Cambridge Bay (-4.31±1.4
cm decade[-1]), Eureka (-4.65±1.7 cm decade[-1]) and Alert (-4.44±1.6 cm decade[-1]) (Table 1). A slight
negative trend is present at Resolute but not statistically significant at the 95% confidence level
(Table 1). Over the 50+ year record, the ice thinned by ~0.24-0.26 m at Cambridge Bay, Eureka
and Alert with essentially negligible change at Resolute. These trends in the CAA are similar to
trends on the Siberian coast (-3.3 cm decade[-1]) [*Polyakov et al.,* 2010] but lower in magnitude
compared to the Barents Sea (-11 cm decade[-1]) [*Gerland et al.,* 2008].

For the shorter record (late 1950s–1989, ~30 years) investigated by BC92 there was a

negative trend at Alert (-7.1 cm decade[-1]), no evidence of a trend at Eureka, and a positive trend at
Resolute (10 cm decade[-1]) but only the positive trend at Resolute was statistically significant at the
95% or greater confidence level. Our results from the present 50+ year record suggest that the
negative trend at Alert is robust and the trend at Eureka is now negative and significant. The trend
at Resolute is now slightly negative however it is not statistically significant.



Typically, ice thickness reaches its maximum in late May with trends toward earlier dates
of maximum ice thickness present at all sites (significant at Resolute, Eureka and Alert; Table 3).
The significant trends are between -2.0±0.1 days decade$^{-1}$ at Eureka to -6.2±1.5 days decade$^{-1}$ at
Resolute. At Resolute, the date of maximum ice thickness is now on average more than a month
earlier than the early 1960's suggesting a shortened growth season although this is not reflected in
the trend in ice thickness. Together, the trends of ice thickness and their recorded dates suggest a
systematic thinning of landfast ice at Cambridge Bay, Eureka and Alert.

**3.3. Ice thickness linkages with snow depth and temperature**
The variability of landfast thickness at these Arctic sites was previously found to be largely
driven by interannual variations in snow depth and air temperature [BC92; *Flato and Brown*,
1996]. With the 50+ year record at the four sites, we can examine the corresponding linkages to
snow depth and temperature which are also summarized in Table 3.
For snow depth, there are positive trends at Eureka and Alert and negative trends at
Cambridge Bay and Resolute (Figure 5). The only trend that is statistically significant at the 95%
confidence is Cambridge Bay at -0.8±0.4 cm decade$^{-1}$ (Table 3). In contrast, BC92 found a
significant positive trend at Alert (4 cm decade$^{-1}$), a trend of low significance in Eureka, and a
negative and significant trend at Resolute (-3.3 cm decade$^{-1}$). Looking at the detrended correlations
(r) between snow depth and ice thickness reveals the strongest correlation at Resolute (r=-0.71)
followed by Eureka (r=-0.66), Alert (r=-0.47) and Cambridge Bay (r=-0.31). While Figure 6
provides evidence from extreme years of the role of deeper snow inhibiting ice growth compared
to thinner snow, the expected statistical correspondence between negative trends in ice thickness
with positive trends in snow depth is only present at Eureka and Alert. This may in part be due to



the single pointwise snow depth and ice thickness measurements made at each point in time, which
fail to capture spatial heterogeneity in the snow depth/ice thickness relationship.

With respect to observed temperature, we find significant warming trends in the spring and

fall at all sites over the 50+ year record (Table 3; Figure 7). Significant warming is also present at
all sites in the summer except Resolute and at all sites during the winter except Eureka (Table 3).
Warming is highest during the fall, at ~0.6°C decade[-1] at all sites (Table 3). The linkage between
temperature and maximum ice thickness weaker than compared to snow depth as only at the
Cambridge Bay site is warming in the spring and winter associated with decreases in maximum
ice thickness with a detrended correlation of ~0.4. This may indicate that temperature plays more
of a role at influencing maximum ice thickness at Cambridge Bay as this site also experienced the
lowest detrended correlation with snow depth (r=-0.31).

Also of interest is that the observed temperature trends over this period differ considerably

than the earlier period investigated in BC92, in which they reported cooling at all the sites, with a
significant cooling trend at Eureka. It was noted that the general cooling over their record coincided
with the 1946-1986 cooling trend over much of the eastern Arctic and northwest Atlantic reported
by *Jones et al.* [1987]. This cooling trend halted during the 1980s and the warming, seen in the
current and longer record, has resumed [*Jones et al.,* 1999]. Arctic land areas have experienced an
overall warming of about ~2°C since the mid-1960s, with area-wide positive temperature
anomalies that show systematic changes since the end of the 20th century, which continued
through 2014 [*Jeffries and Richter-Menge*, 2015]. Recently, warming in Canadian Arctic regions
was found to be greater than the pan-Arctic trend by up to 0.2°C decade[-1] [*Tivy et al.,* 2011].

**4. Results and Discussion: Models**



### 4.1. Climatology

In order to compare seasonal cycles and trends in landfast ice thickness and snow depth between models and observations, we limit our comparison to models with a reasonable representation of the CAA, i.e. those with an open Parry Channel (i.e. bcc-csm-1-1, bcc-csm-1-1m, CNRM-CM5, ACCESS1-0, ACCESS1-3, FIO-ESM, EC-EARTH, inmcm4, MIROC5, MPI-ESM-LR, MPI-ESM-MR, MRI-CGCM3, CCSM4, NorESM1-M, NorESM1-ME, GFDL-CM3, GFDL-ESM2G, GFL-ESM2M, CESM1-BCG, CESM1-CAM5, CESM-WACCM). In these models, sufficient spatial resolution allows us to find sample points that are almost collocated to *in situ* observation locations. The sample points were determined by finding the closest ocean grid point where the sea ice is packed for a good portion of year but not all year. Grid points with this characteristic therefore share the most important feature of the landfast ice at our observations locations: it is not perennial. Mathematically, we sought sample points where the sea ice concentration is on average above 85% for more than one month but less than 11 months over the 1955-2014 period. The Eureka site is however particularly challenging for models because it lies deep in a very narrow channel, which is only resolved by the MPI-ESM-MR in the CMIP5. As a result, for most models, the sample point for Eureka is located on the western shore of Ellesmere Island.

The seasonal cycle (1955-2014) of median ice thickness from CMIP5 (black), ORA-IP models CGLORS, ORAP5.0 and GLORYS2V3 (blue), ECCO-v4 (green) and UR025.4 (red) is shown in Figure 8. ORA-IP models have been split into three groups based, respectively, on their high, medium and low ice thicknesses at Alert. Ice thickness from CMIP5 is comparable to observations (Figure 2) at Cambridge Bay and Resolute with maximum ice thickness reaching 200 cm. The ORA-IP models are less consistent. ECCO-v4 tends to have thicker sea ice than



observations at Cambridge Bay, Resolute and Eureka but thinner at Alert. CGLORS, ORAP5.0,
and GLORYS2V3, on the other hand, are comparable to observations at Cambridge Bay, Resolute
and Eureka but have extremely thick and perennial ice close to Alert.

The seasonal cycle (1955-2014) of median snow depth from CMIP5 is shown in Figure 9.

CMIP5 models indicate a linear increase similar to observations reaching a maximum of ~20 cm
in April or May. This is lower than the observed maximum at Resolute, Eureka and Alert but is
about twice as much as at Cambridge Bay. While the snow depth reaches zero during the summer
at Eureka and Alert in models, the sea ice thickness does not (Figure 8), unlike in observations.
This likely reflects the fact that thick, mobile ice is located in the vicinity of these sample points
in models. The seasonal cycle over packed ice in these models thus gives a reasonable
representation of the seasonal cycle over landfast ice in the CAA, especially in the southern region
of the CAA. Overall, this comparison shows how recent improvements in sea ice model resolution
allows comparisons with observations that required dynamical downscaling techniques in the
previous generation of sea ice models [i.e. *Dumas et al. 2005; Sou and Flato, 2013*].

Despite relatively high spatial resolution, PIOMAS does not resolve seasonal ice thickness

along the coasts and within the very narrow channels within the CAA (not shown). As a result,
Cambridge Bay and Resolute Bay sites represent the only long-term monitoring sites within the
CAA suitable for comparison since PIOMAS. The monthly time series of PIOMAS ice and snow
thickness estimates at Cambridge Bay and Resolute is shown in Figure 10. The seasonal cycle of
ice growth at Cambridge Bay and Resolute is representative compared to observations (Figure 2)
but PIOMAS estimates retain more ice in August and September, particularly at Resolute. Ice
growth reaches a maximum in April at Cambridge and in May at Resolute which is 1-month earlier
compared to observations. Snow depth follows a linear increase similar to observations (Figure 3)



with good agreement at Cambridge Bay but considerably underestimates snow depth at Resolute
(Figure 10). *Schweiger et al.* [2011] performed a detailed comparison of PIOMAS ice thickness
values against *in situ* and Ice, Cloud, and land Elevation Satellite (ICESat) ice thickness
observations and found strong correlations. They determined a root mean square error (RMSE) of
~0.76 m and noted that PIOMAS generally overestimates thinner ice and underestimates thicker
ice. At both sites within the CAA, PIOMAS ice thickness data is in reasonably good agreement
with *in situ* observations with RMSE's of 0.29 cm at Cambridge Bay and 0.68 cm at Resolute
(Figure 11). The systematic overestimate of thinner ice reported by *Schweiger et al.* [2011] is more
apparent at Resolute than Cambridge Bay (Figure 11). The  higher latitude regions of the CAA
where there is an intricate mix of seasonal first-year ice and multi-year ice is a problem for
PIOMAS and thus contributes to the larger discrepancy at Resolute compared to Cambridge Bay.

**4.2. Trends**

The spatial distribution of maximum sea ice thickness trends from ORA-IP and CMIP5 is

illustrated in Figures 12. It is particularly apparent that the high resolution models exhibit a similar
North-South trend pattern as for the observational stations (Figure 2), albeit with overestimated
negative thickness trends. The general pattern and magnitude of the thickness trends are roughly
in accordance with the temperature trends in these models (not shown). One exception is the ORA-
IP CGLORS that have positive thickness trends (Figure 12a). This is robust and it appears that the
model is not completely equilibrated in the CAA and exhibit large month-to-month adjustments.
Model ORAP5.0 also is not completely equilibrated in the region for years 1979-1984. During
those years, it exhibits large inter annual changes in thickness. For this reason, we are only
considering years 1985-2013 for this model.



For PIOMAS, the North-South overestimated trend is also present (not shown) as with
CMIP5 and ORA-IP. Looking specifically at trends near the observed sites indicates that the mean
maximum ice thickness linear trend from at Cambridge Bay is -13.4+3.4 cm decade$^{-1}$ which is
almost double the observational trend of 6.2±2.4 cm decade$^{-1}$. At Resolute, the PIOMAS linear
trend is 24.0±4.1 cm decade$^{-1}$ which is considerably stronger than the observational trend of -
4.9±3.51 cm decade$^{-1}$.

**4.3. Ice thickness linkages with snow depth and temperature**
Even though ORA-IP models have unrealistically large thickness trends, the pattern of inter
annual correlation (detrended) between winter temperatures and thicknesses is roughly consistent
across models (Figure 13). Some ORA-IP models also experience positive correlations (e.g.
CGLORS, ORAP5.0, GLORYS2V3 and UR025.4) that are mostly located north of the CAA or
within the CAA in regions where multi-year ice is known to be present.  It is possible that warmer
temperatures are associated with an increased flux of thicker multi-year ice into the CAA which is
known to occur [e.g. *Howell et al.,* 2013] but the driving processes responsible for these positive
correlations require more investigation. In CMIP5 models, no model exhibits positive correlations
with temperature that resemble ORA-IP models over the CAA. Although the time series for the
ORA-IP models is short and the positive correlations are not statistically significant, this behavior
suggest that care should be taken when using these ORA-IP models to study the interannual
variability in the Canadian Arctic.
In the CMIP5 models, significant winter snow depth trends are more strongly negative in
the North than in the South (Figure 14). This is in disagreement with point observations presented
in the previous sections that showed slightly positive snow depth trends at Alert and negative



trends at Cambridge Bay. Although only based on limited point *in situ* observations, this suggests
that over the last decades winter precipitation at Alert increased faster than warming temperature
could increase melting, a compensation that is clearly not captured in CMIP5 models.

**5. Conclusions**

Over the 50+ year in situ observational record, negative trends in maximum (end-of-winter)

ice thickness are found at all four sites with statistically significant trends present at Cambridge
Bay, Eureka and Alert. Negative trends in the day of maximum ice thickness are also present at all
sites and statistically significant at Resolute, Eureka and Alert. Together, these trends suggest
thinning of landfast ice in the CAA, where little evidence was found in the shorter record analyzed
in an earlier study (BC92). Even though warming is seen at all sites, changes in ice thickness is
also attributable to variability in snow depth, which plays a dominant role in controlling the
interannual mean and variability of ice thickness. Within the CAA, increases in snow depth are
contributing to decreased trends in maximum ice thickness at Eureka and Alert but thus far appear
to be exerting less of an impact on maximum ice thickness at Resolute and Cambridge Bay. Freeze
onset at these sites is increasing at ~3-6 days decade[-1] [*Howell et al.,* 2009] and the delayed ice
formation could play more of a role at the in the southern sites because of a longer open water
season.

Comparison of CMIP5, ORA-IP and PIOMAS simulations with observations indicate a

reasonable representation of the landfast ice thickness monthly climatology within the CAA. This
is particularly apparent when seasonal first-year ice dominates the icescape (i.e. Cambridge Bay).
Despite improvements in spatial resolution, mixed ice types (i.e. seasonal and multi-year) present
at the sub-grid cell resolution are likely problems for model estimates within the CAA. The overall



thickness of ice within the CAA in the current generation of models is too high. As a result, trends
are unrealistic and far exceed observations (by upwards of -50 cm decade$^{-1}$) in part because the
initial ice thickness is too large. The problem is particularly acute in the ORA-IP models where
large and unrealistic inter annual changes in thickness suggest that the models are not fully
equilibrated.

Over the mobile Arctic Ocean ice cover, the combined record of submarine and ICESat

thickness estimates suggest that winter sea ice thickness in the central Arctic has thinned from 3.64
m in 1980 to 1.75 m by 2009 [*Rothrock et al.,* 2008; *Kwok and Rothrock*, 2009] – a linear rate of
over -60 cm decade$^{-1}$ that is mostly due to the loss of multi-year ice. However, the contribution of
seasonal ice to that rate is not available. As seasonal ice, becomes the dominant ice type, the focus
has shifted to understanding the behavior of seasonal ice thickness. Between 1991 and 2003,
*Melling et al.* [2005] found only a small trend (-7 cm decade$^{-1}$), though of low statistical
significance, in the seasonal pack in the Beaufort Sea. In the short ICESat record of ice thickness
(2003-2008), *Kwok et al.* [2009] also found negligible trend in the seasonal ice cover. This led
them to speculate that a thinner snow cover during to the later start of the growth season is
conducive to higher ice production as a result of reduced accumulation of that large fraction of
snow that typically falls in October and November. However, over the seasonal ice cover there is
the additional contribution of ice deformation on the mean of the thickness distribution.

While the impact of the snow cover on ice thickness is well known, the significant

correlations at Resolute, Eureka and Alert suggest that the higher sensitivity to changes in snow
depth could easily mask the warming signal on both fast and offshore ice. The dependency between
ice thickness trends and warming trends is only weakly present at Cambridge Bay (r=0.4) and
further points out the dominance of snow depth because of the large variability of the thickness



trends compared to the relatively low scatter in the temperature trends. Thus, even in this limited
data set, we can see the dominant role played by snow depth in determining the interannual
variability of the maximum landfast ice thickness. This again highlights that the primary factor is
the amount and timing of snow accumulation, not air temperature. However, it is worth noting that
few of the current generation models show coherent relationships between ice thickness, snow
depth and temperature over the longer term record.

**Authors Contributions**
S.E.L.H, F.L and R.K designed the study, performed the analysis and wrote the manuscript with
input from C.D. and J.K.

**Acknowledgements**
The authors with to thank all the individuals responsible for collecting landfast ice and snow
thickness measurements in the Canadian Arctic over the past 50+ years.

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



Table 1. CMIP5 models used in this study, the number of realizations with ice data and the
number of realizations with sea ice transport data

| | w/ ice | | w/ ice |
|---|---|---|---|
| bcc-csm1-1 | 1 | MIROC-ESM-CHEM | 1 |
| bcc-csm1-1-m | 1 | MIROC5 | 3 |
| BNU-ESM | 1 | HadGEM2-CC | 1 |
| CanESM2 | 5 | HadGEM2-ES | 4 |
| CMCC-CESM | 1 | MPI-ESM-LR | 3 |
| CMCC-CM | 1 | MPI-ESM-MR | 1 |
| CMCC-CMS | 1 | MRI-CGCM3 | 1 |
| CNRM-CM5 | 5 | CCSM4 | 6 |
| ACCESS1.0 | 1 | NorESM1-M | 1 |
| ACCESS1.3 | 1 | NorESM1-ME | 1 |
| CSIRO-Mk3.6.0 | 10 | GFDL-CM3 | 1 |
| FIO-ESM | 1 | GFDL-ESM2G | 1 |
| EC-EARTH | 6 | GFDL-ESM2M | 1 |
| inmcm4 | 1 | CESM1(BGC) | 1 |
| FGOALS-g2 | 1 | CESM1(CAM5) | 3 |
| MIROC-ESM | 1 | CESM1(WACCM) | 3 |




Table 2. Summary of ORA-IP models characteristics

| Model Name | CGLORS | ECCO-v4 | GLORYS2V3 | ORAP5.0 | UR025.4 |
|---|---|---|---|---|---|
| Institute | CMCC | JPL-NASA-MIT-AER | Mercator Océan | ECMWF | University of Reading |
| Resolution | ORCA0.25° | ~40km in the Arctic | ORCA0.25° | ORCA0.25° | ORCA0.25° |
| Ocean Model | NEMO 3.2.1 | MITgcm | NEMO 3.1 | NEMO3.4 | NEMO 3.2 |
| Sea ice Model | LIM2 | MITgcm | LIM2 (with EVP rheology) | LIM2 | LIM2 |
| Time period considered | 1982-2012 | 1991-2011 | 1993-2013 | 1985-2013 | 1993-2010 |
| Atmospheric forcing | ERA-Interim | ERA-Interim | ERA-Interim | ERA-Interim | ERA-Interim |
| Sea ice product assimilated | NSIDC NASA-Team Daily | NSIDC Bootstrap Monthly | IFREMER/CERSAT | NOAA / OSTIA combination | EUMETSAT OSI-SAF |






Table 3. Observed maximum ice thickness, snow depth, and surface air temperature at four
landfast ice sites in the Canadian Arctic Archipelago. The bold text indicates statistical
significance of the linear trend at 95% or greater.

| | Cambridge Bay | Resolute | Eureka | Alert |
|---|---|---|---|---|
| Period | 1960-2014 | 1957-2014 | 1957-2014 | 1957-2014 |
| Ice Thickness, $h_{ice}$ | | | | |
| Mean of *max* $h_{ice}$ (m) | 2.11±0.19 | 2.02±0.19 | 2.27±0.23 | 1.98±0.22 |
| Trend of *max* $h_{ice}$ (cm decade$^{-1}$) | **-4.31±1.4** | -0.5±1.6 | **-4.65±1.7** | **-4.44±1.6** |
| Day of *max* $h_{ice}$ | 24 May±17 | 25 May±21 | 26 May±12 | 27 May±16 |
| Trend of day of *max* $h_{ice}$ (days decade$^{-1}$) | -0.87±1.5 | **-6.2±1.5** | **-2.0±0.1** | **-3.0±1.2** |
| | | | | |
| Snow depth ($h_{snow}$ ) | | | | |
| Mean Oct-May $h_{snow}$ (cm) | 8.4±4.2 | 22.6±10 | 17.6±5.8 | 18.4±6.2 |
| Trend of Oct-May $h_{snow}$ (cm decade$^{-1}$) | **-0.8±0.4** | -0.75±0.8 | 0.54±0.5 | 0.26±0.5 |
| | | | | |
| Temperature | | | | |
| Winter (Dec-Feb) Mean (°C) | -31.3±2.0 | -30.8±1.9 | -36.0±2.0 | -31.2±1.6 |
| Winter (Dec-Feb) (°C/decade) | **0.59±0.2** | **0.35±0.1** | 0.23±0.2 | **0.38±0.1** |
| Spring (Mar-May) Mean (°C) | -20.0±1.8 | -21.1±1.8 | -24.9±2.0 | -22.8±1.8 |
| Spring (Mar-May) (°C/decade) | **0.47±0.1** | **0.57±0.1** | **0.44±0.1** | **0.32±0.1** |
| Summer (Jun-Aug) Mean (°C) | 5.9±1.4 | 2.3±1.3 | 3.9±1.2 | 1.3±0.8 |
| Summer (Jun-Aug) (°C/decade) | **0.30±0.1** | 0.17±0.2 | **0.21±0.1** | **0.1±0.1** |
| Fall (Sep-Nov) Mean (°C) | -11.1±2.0 | -13.8±2.0 | -19.6±2.2 | -18.0±1.7 |
| Fall (Sep-Nov) (°C/decade) | **0.60±0.2** | **0.67±0.1** | **0.68±0.2** | **0.56±0.1** |




**List of Figures**



Figure 14. Same as Figure 12f but for snow depth trends (ONDFJMAM).





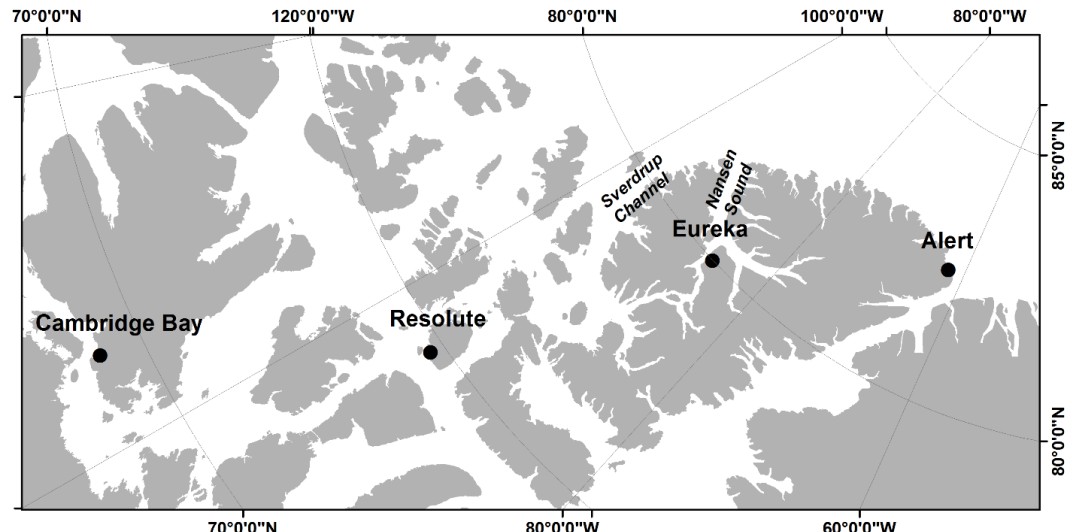

Figure 1. Map of the central Canadian Arctic Archipelago showing the location of the landfast
snow and thickness observations.





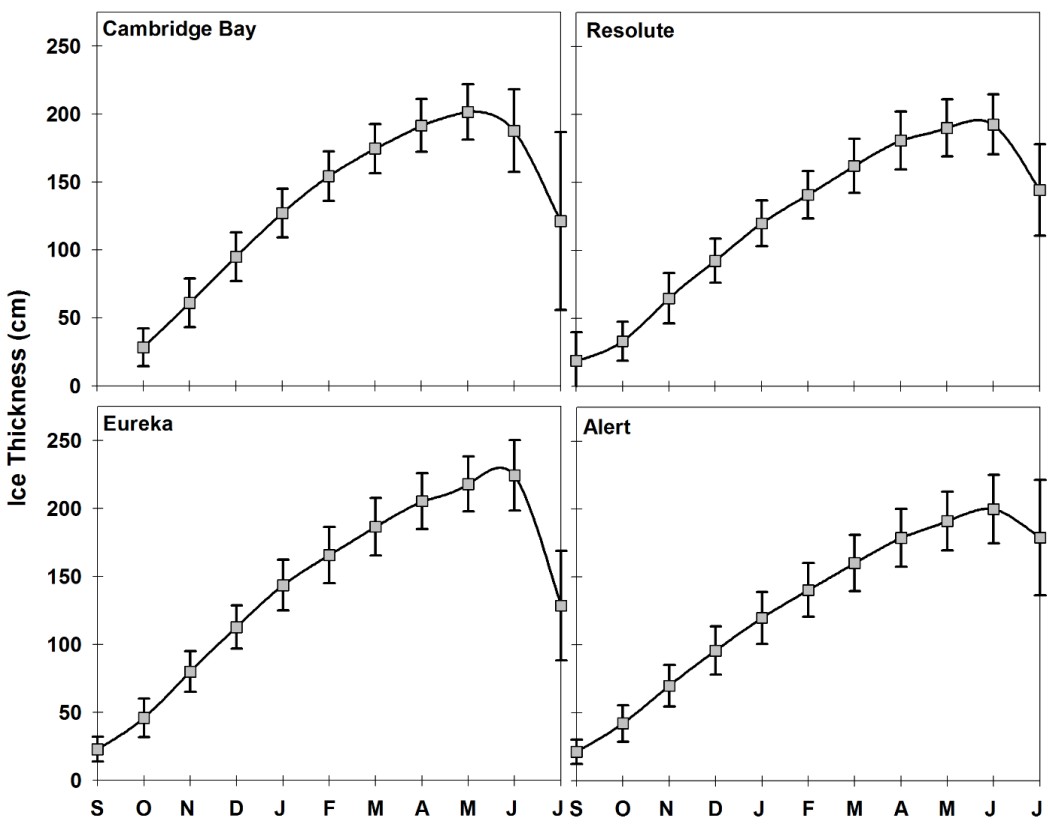

Figure 2. Seasonal cycle of observed mean ice thickness at the four sites (1960-2014).





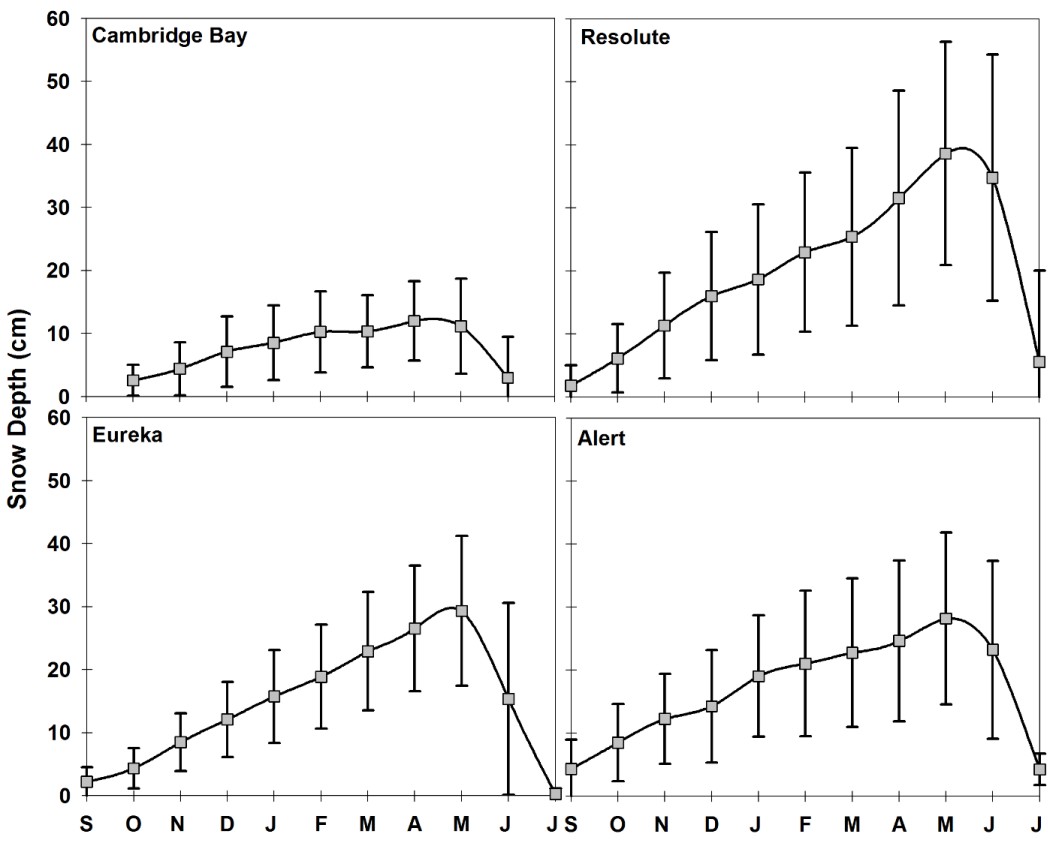

Figure 3. Seasonal cycle of observed mean snow depth at the four sites (1960-2014).





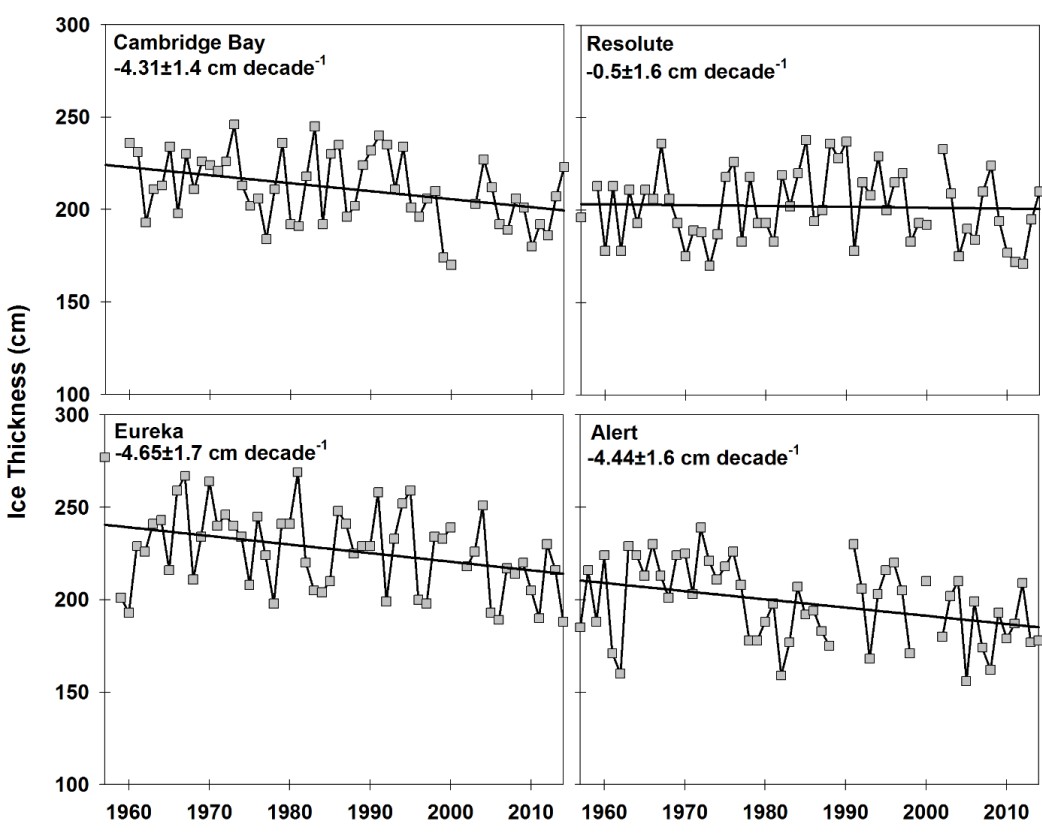

Figure 4. Time series and trend of observed maximum ice thickness at the four sites.



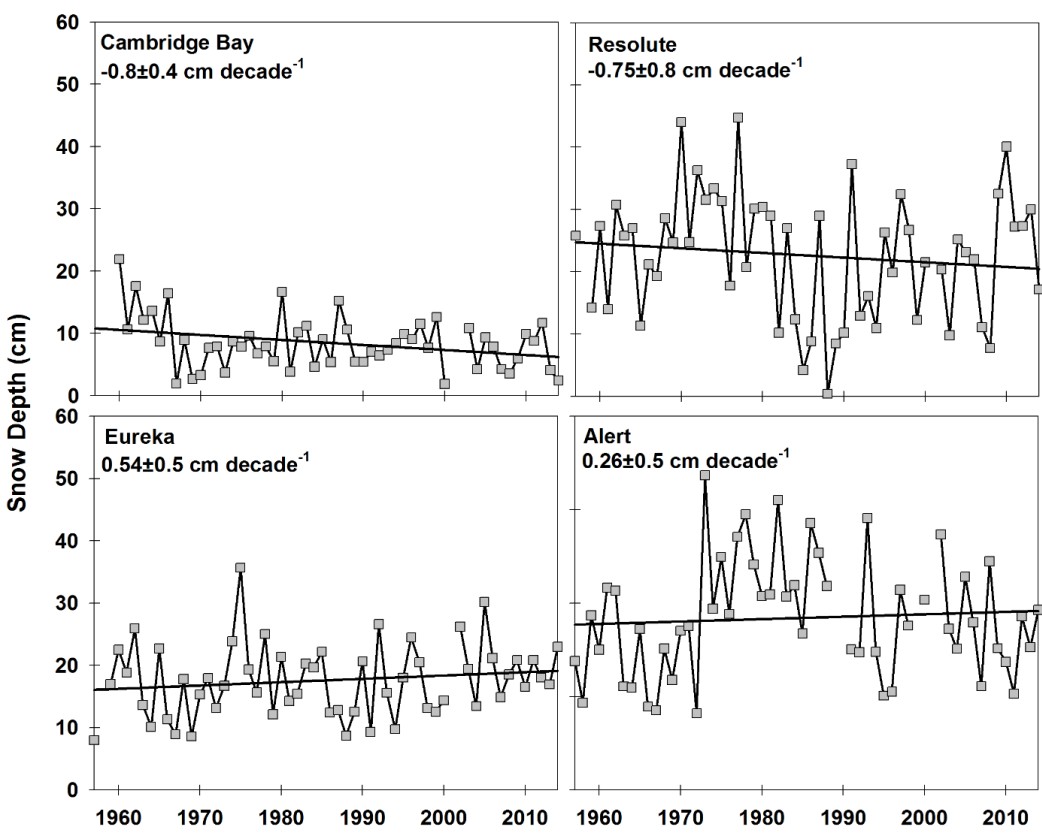

Figure 5. Time series and trend of observed mean October through May snow depth at the four
sites.





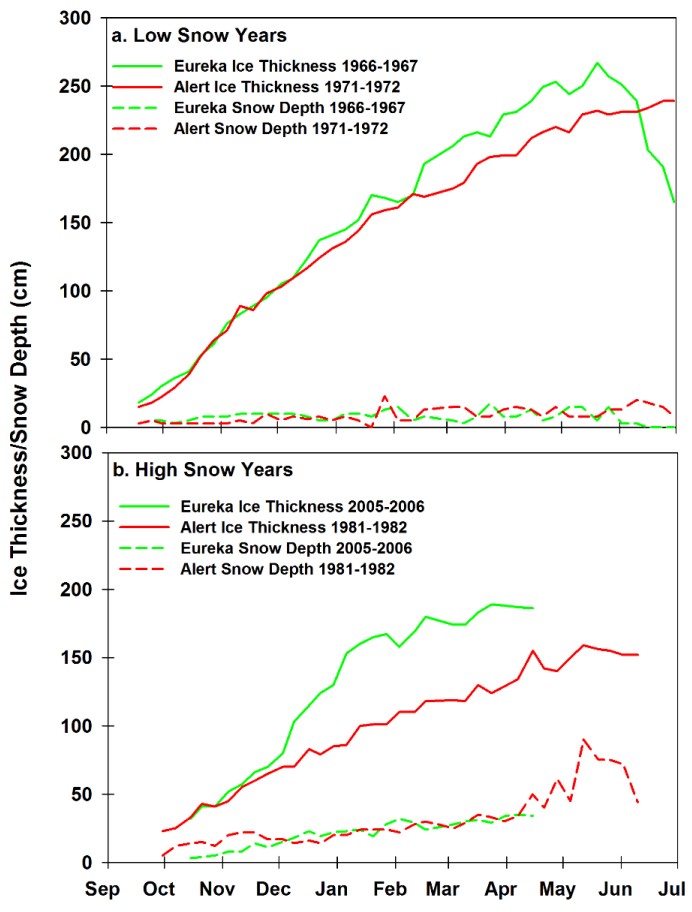

Figure 6. Weekly time series of ice thickness and snow depth at Eureka and Alert for (a) low
snow years and (b) high snow years.



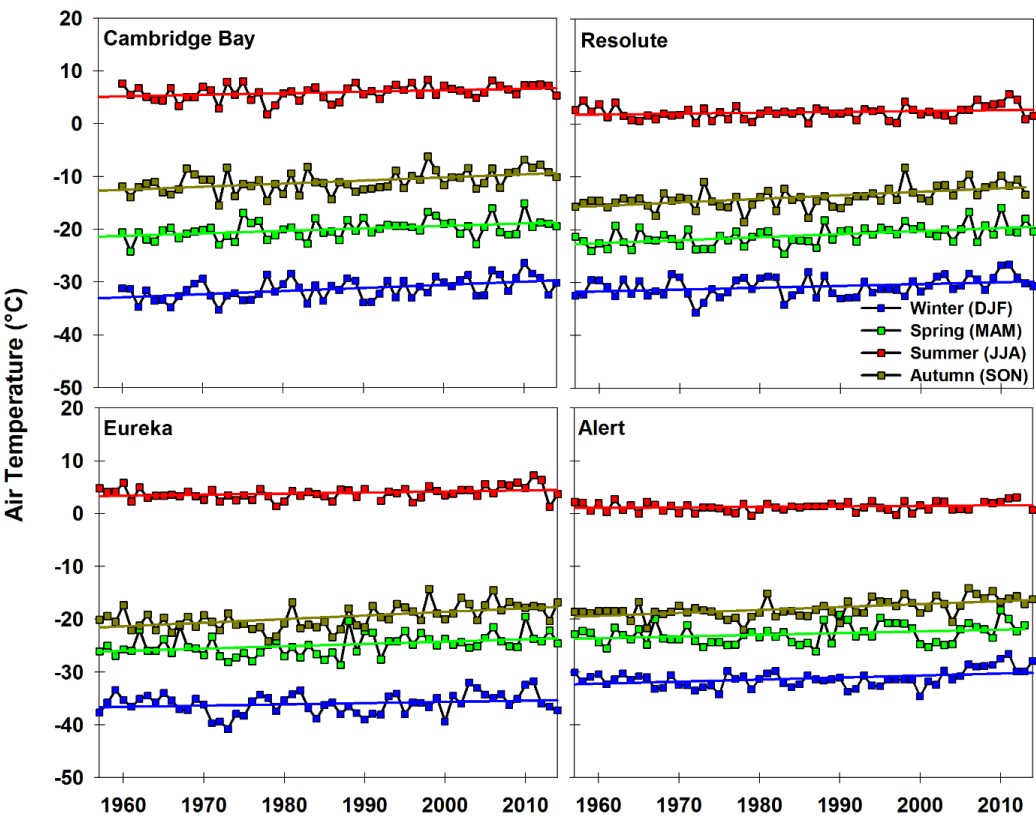

Figure 7. Time series of mean air temperature during winter (DFJ), spring, (MAM), summer
(JJA) and autumn (SON) at the four sites.



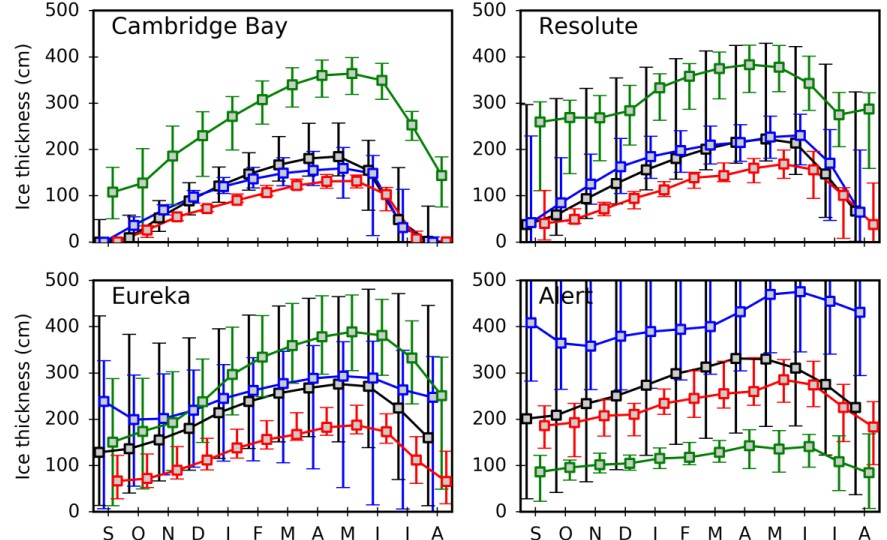

Figure 8. CMIP5 median sea ice thickness seasonal cycle and evolution (1955-2014) at stations
(black). Median of ORA-IP models CGLORS, ORAP5.0 and GLORYS2V3 (blue), ECCO-v4
(green) and UR025.4 (red). Whiskers indicate the 5th and 95th percentiles.



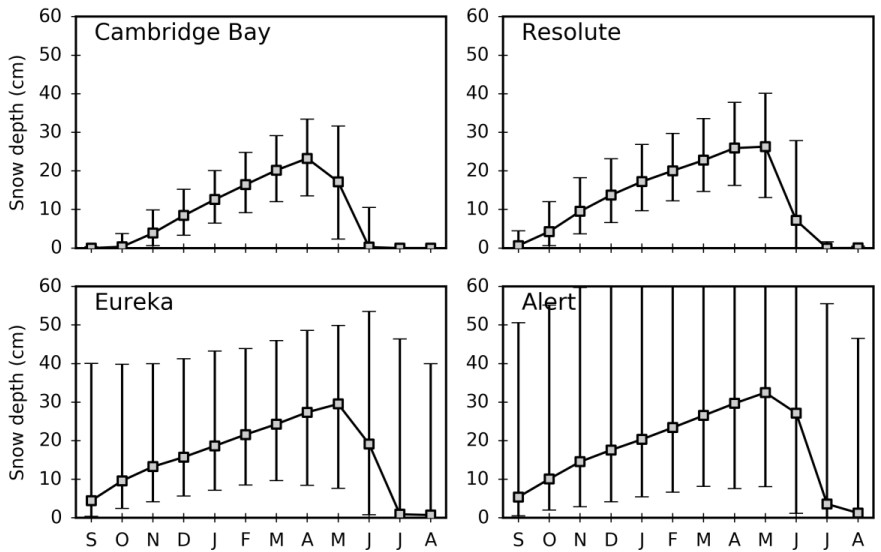

Figure 9. Same as Figure 10 for snow depth and only for CMIP5 models.





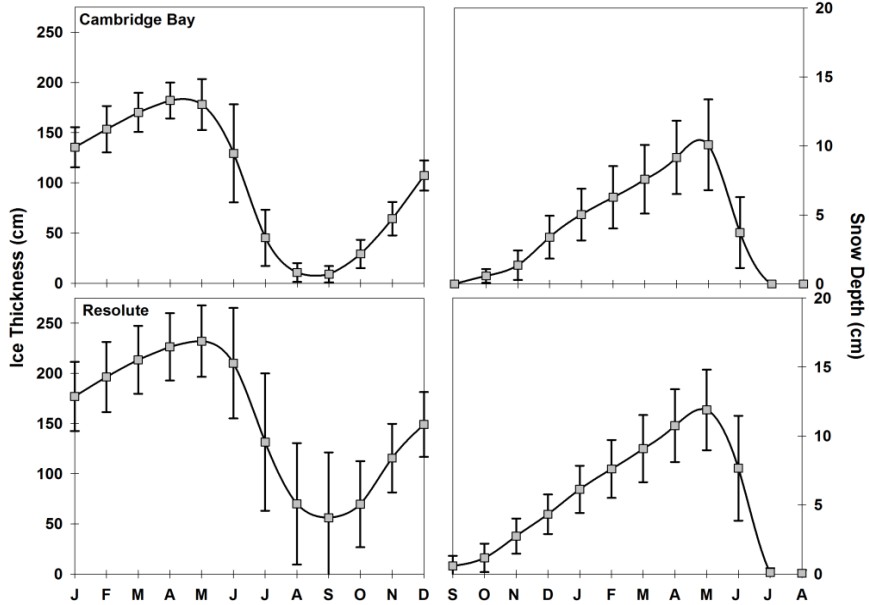

Figure 10. Seasonal cycle of observed mean ice thickness (left) and snow depth (right) from
PIOMAS at Cambridge Bay and Resolute (1979-2014).





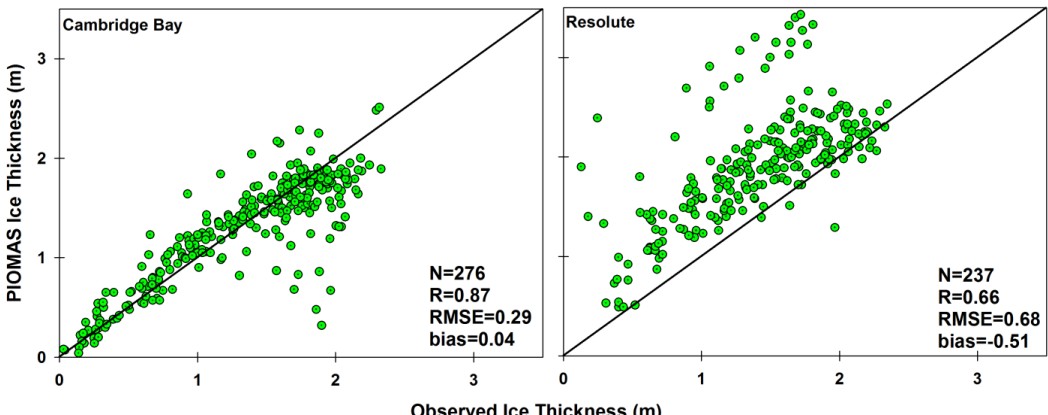

Figure 11. Comparison of PIOMAS ice thickness with ice thickness observations from
Environment Canada's ice thickness monitoring sites at Cambridge Bay and Resolute. The data
covers the period 1979-2014.





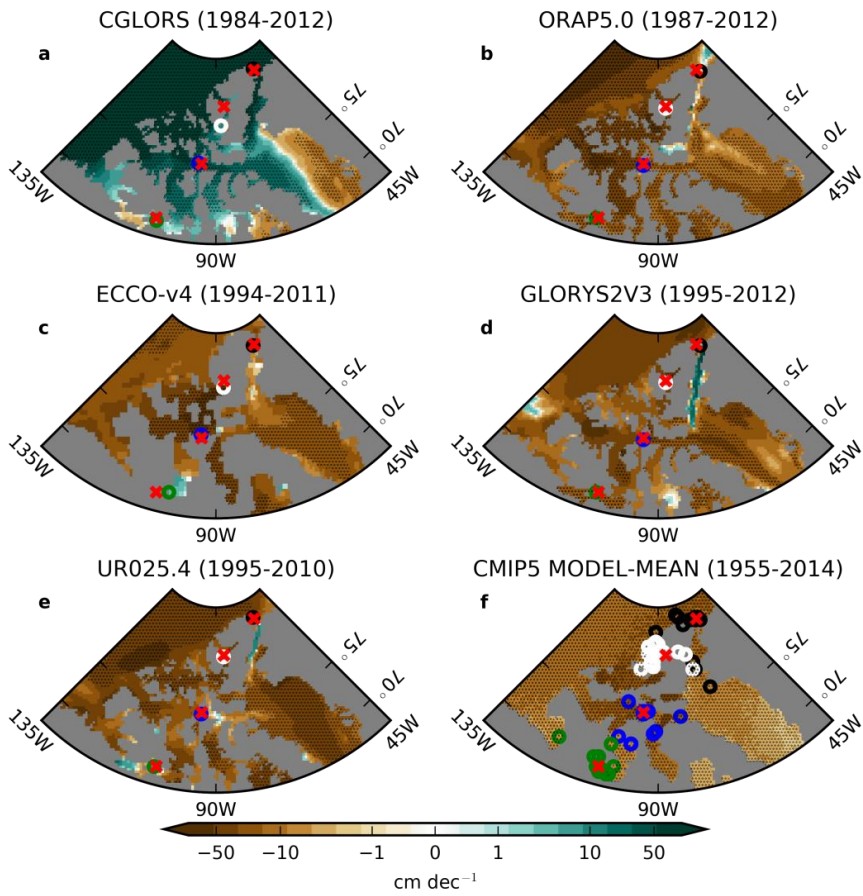

Figure 12. **a-e:** Maximum sea ice thickness trends in ORA-IP simulations. **f:** Same for CMIP5
MODEL-MEAN. From South to North, o's indicate Cambridge Bay (green), Resolute (blue),
Eureka (white) and Alert (black) and x's indicate the corresponding measurement stations. In f,
one o per model is shown." The stippling indicates p-values less than 0.05, corrected using the
False Discovery Rate (FDR) method with a global pFDR-values less than 0.10 [*Wilks*, 2006].
The colorbar is linear from -10 cm dec$^{-1}$ to 10 cm dec$^{-1}$ and symmetric logarithmic beyond these
values.





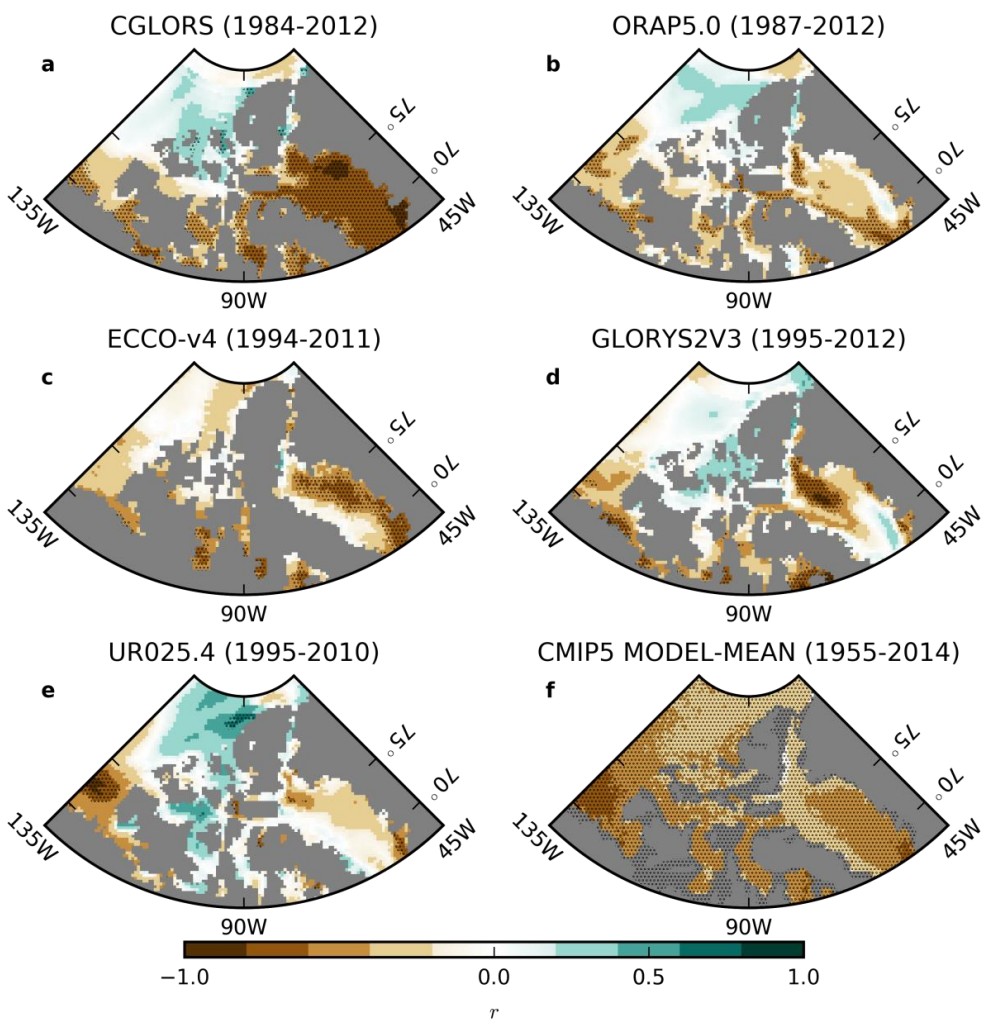

Figure 13. **a-e:** Pearson correlation of detrended maximum sea ice thickness in ORA-IP with detrended ONDJFMAM ERA-INTERIM 2m temperature. **f:** Same but for CMIP5 MODEL-MEAN. The stippling indicates p-values less than 0.05, corrected using the False Discovery Rate (FDR) method with a global pFDR-values less than 0.10 [*Wilks*, 2006].





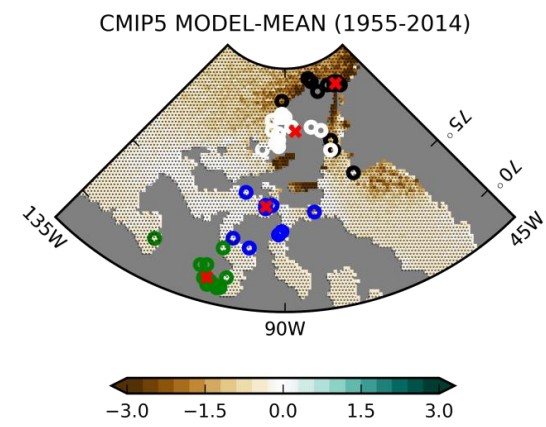

Figure 14. Same as Figure 12**f** but for snow depth trends (ONDFJMAM).