# Peer review of "Stephen. E. L. Howell1, Frédéric Laliberté1, Ron Kwok2, Chris Derksen1 and Joshua King1"

_The Cryosphere, 2016_

## Referee Comment (RC1) · D. Bailey (Referee) · 29 Apr 2016

This manuscript describes an analysis of ice thickness and snow depth from a combination of local observations and from CMIP models. Overall, I believe this article was well written and would be of interest to readers of the Cryosphere journal. I do have one minor suggestion for the authors. Instead of trying to pick model points closest to the observed sites, perhaps a better approach is to average over the whole Canadian Arctic Archipelago. As the authors point out, the CMIP models do not have the resolution to properly represent the channels in this area. So, a larger area average might be more meaningful. This could be compared to the average of the four observed locations. Also, perhaps some thoughts on how the relationships between land fast ice and snow depth could be better represented in the models would be helpful.

---

## Referee Comment (RC2) · Anonymous Referee #2 · 2 May 2016

**1   General comments**

This paper presents an analysis of time series of land-fast ice thickness and related variables from four locations in the Canadian Arctic Archipelago (CAA), obtained from both observations and models. The observations presented are very interesting, giving potentially a unique view of the evolution of ice thickness in the CAA, as well as some insight into the climate and climate change in the CAA. The data, results, and conclusions are not ground breaking or spectacular, but can serve as a solid addition to our knowledge of the Arctic. The authors also use the observational data to evaluate model results. This is a useful exercise, but still needs some work.

The paper is mostly clear and concise. The authors' approach in working with the data

and models is also good in general. There are a number of points that need to be addressed before I can recommend publication, but I am confident that the authors can address those in a satisfactory manner. Given that I expect the resulting paper to become a good, relatively low-key, but solid addition to our understanding of the CAA climate.

My main concern regarding this paper is with the conclusions the authors draw at the end. These are too often poorly supported by the data, or even not at all. In some cases the discussion is lacking so that it is not clear what conclusions to draw. The most confusing aspect is the role of snow and temperature, where it is not always clear when we should be thinking about de-trended correlations or correlation between trends. It is also not clear how you calculate the correlation between maximum ice thickness and temperature and it's not immediately clear how this should be done (do you want to use a mean temperature over the growing season for instance?)

In particular, in the first paragraph of the conclusions the authors state that "[e]ven though warming is seen at all sites, changes in ice thickness is also attributable to variability in snow depth, which plays a dominant role in controlling the inter-annual mean and variability of ice thickness". Here the authors appear to be mixing the long-term trend in temperature (the warming) with the inter-annual variability, which they show is highly correlated with snow thickness (the de-trended correlation is correctly used for this). It seems to me that the inter-annual variability in maximum ice thickness is controlled by variability in snow thickness and that the long-term trend is caused by a long term trend in temperature. But I'll come back to this point below.

The authors then go on to say "[w]ithin the CAA, increases in snow depth are contributing to decreased trends in maximum ice thickness at Eureka and Alert but this far appear to be exerting less of an impact on maximum ice thickness at Resolute and Cambridge Bay". I don't understand how the authors reach this conclusion. The trend in snow thickness is only significant at Cambridge Bay so we can safely assume that only there can a trend in snow thickness contribute to a trend in ice thickness.

Turning to the relationship between ice thickness and temperature, in paragraph four of the conclusions the authors state: "the significant correlations at Resolute, Eureka and Alert suggest that the higher sensitivity to changes in snow depth could easily mask the warming signal on both fast and offshore ice". I don't really feel this has been shown in the paper. I very much expect this statement to be true, but it needs to be better argued.

The authors then go on to say "[t]he dependency between ice thickness trends and warming trends is only weakly present at Cambridge Bay ($r = 0.4$)", but in the text it is clearly stated that the given $r$-value is related to the de-trended correlation and has as such nothing to do with the thickness and warming trends.

My conclusions, after reading the paper (and not your conclusions) are that

- Snow thickness is the main driver of inter-annual variations in maximum ice thickness (high de-trended correlation at two out of three sites, and reasonably high correlation at the other two).

- Inter-annual variations in air temperature are only weakly correlated with maximum ice thickness (max $r = 0.4$).

- A trend in air temperature is the main driver of the trend in maximum ice thickness. This is because there is a significant warming trend at all sites, but only significant snow thickness trend at one.

In short the inter-annual variations in maximum ice thickness are caused by variations in snow, while the long term trend is caused by a long term trend in temperature. The fact that there is no ice thickness trend at Resolute is probably related to the timing of snow fall, but this is only mentioned once in the paper and not explored in any depth. The authors conclude that change in snow thickness, not temperature is the driver of both inter-annual and long-term changes, but I can only agree with the former part of that statement, not the latter.

Now, I don't have access to all the data and haven't spent much time on the analysis, so maybe I'm completely wrong and the authors right. But if this is the case then the authors need to make a more convincing point in their presentation of the results and conclusions.

In addition I'm also concerned over sections 4.2, and 4.3. First of all, the ORA-IP models are all run over a different, and much shorter time period than is covered by the observations. I therefore doubt that it is appropriate to use those to calculate trends that should be compared to the observed trends. In addition the first paragraph of that section left me very confused. You propose comparing figures 2 and 12, but the former shows thickness seasonal cycle, while the latter shows thickness trends. You talk about a general pattern, but this is too vague and qualitative for my taste. Also, I'm not sure there is a north-south pattern in the observed trends, we can at least not see this from table 2. In short, it seems to me you should focus on CMIP5 for the trend analysis because the ORA-IP periods are too short. It could be OK to use the longest runs, i.e. CGLORS, ORAP5.0, and PIOMAS (that runs from 1979, right?), but be careful.

**2 Specific comments**

**l. 25** There is hardly a need for a reference to what land-fast ice is, but if you want one I would recommend the WMO definition. On the other hand, I would like to see a reference for the statement "this ice typically extends to the 20–30 m isobath". It is right, but should be backed up by some references (I think Hajo Eicken recently published a nice overview and you already have Mahoney et al. in your reference list).

**l. 101** Here you say that the CMIP5 runs extend from 1980 to 2099, but later you say that you use the period 1955–2014. But doesn't the historical run go back to 1850?

**l. 117** I would add PIOMAS to table 2 for completeness.

**l. 162** You can do better than suggesting a shortened growth season. In the conclusions you cite Howell et al. who show a later onset of melt and if you combine this with your results you are conclusively demonstrating a shortened growth season.

**l. 171** You should not tell us which trends there are, only to then tell us that they are all insignificant except one. In fact you should only discuss the significant trends and if you want to mention the others then don't spend to much time on them.

**l. 179** Here you say that there is statistical correspondence between the ice and snow thickness trends at Eureka and Alert, but the snow thickness trends there are not significant. This does not make sense to me. Also, it is not clear what you mean by "statistical correspondence".

**l. 186** I couldn't understand the sentence starting on this line. Please rewrite.

**l. 207** For completeness you should note Parry Channel in figure 1. But I'm not sure about this condition, since ocean and ice dynamics presumably play a small role here. If you want a more quantitative measure than "a reasonable representation of the CAA" then I would suggest putting a limit on the resolution in the CAA.

**l. 215** Picking single points to analyse is always a bit questionable. You should make sure, and state that the point selection does not affect your results, as long as the points selected follow your selection criterion. I suppose that there were more than one candidates for each model and location. Also, some of the model points are quite far away from the observation station. I noted especially points representing Alert and Cambridge Bay which are so far away from the stations that I would expect them to belong to a different atmospheric and oceanographic regime (see figure 12f).

**l. 221** Here you say that the seasonal cycle is calculated over 1955–2014, but this does not fit your earlier statement about the CMIP5 models or the information in table 2.

**l. 235** It is not clear to me why this statement should be true. Please elaborate.

**l. 264** Which models are the "high resolution models"?

**l. 276** In table 2 the trend is $4.3 \pm 1$, not $6.2 \pm 2.4$. There is also an inconsistency in signs in this paragraph.

**l. 289** If the trends are not significant then we can't learn much from them.

**l. 295** Again, trends that are not significant don't warrant much of a discussion.

**l. 301** Don't talk about the insignificant trends as if they matter.

**l. 324** This paragraph doesn't belong in your conclusions since it contains nothing from the current work.

**3  Technical corrections**

**l. 20** Replace "two magnitudes" with "two orders of magnitude".

**l. 131** Replace "2.27 m that is likely ..." with "2.27 m, which is likely ...'

**l. 169** Here it sounds as if it's the linkages that are summarised in table 3, but this is not the case.

**l. 176** The r should be italicised and the equations would look much nicer if the were within dollar signs (assuming you're using latex). I.e. $r = -0.66$, instead of r=-0.66.

**l. 193** Replace "than the earlier period" with "from the earlier period".

**l. 213** Replace "good portion of year but not all year" with "good portion of the year, but not all year".

---

## Referee Comment (RC3) · Anonymous Referee #2 · 2 May 2016

In my review I forgot to include the comments I'd written down w.r.t. the figures. These are:

Figure 7: I would think that having separate panes for each season would make more sense. There should be much less difference between the temperature at different locations than there is between the different seasons. That way we can more easily see the trend. As it is all the lines look very flat.

Figure 8: I would like to see the observations on these figures as well. They may become a little crowded, I'll admit, but it's important to be able to see, at a glance how the models depart from observations.

Figures 9 & 10: The observations should be shown here as well.

Figure 14: You say "same as figure 12f", but I just looked at the previous figure, no. 13 (without thinking) and got very confused. I think it would be better to spell out what's happening in figure 14.

---

## Author Comment (AC1) · 2 Jun 2016

1   General comments

This paper presents an analysis of time series of land-fast ice thickness and related variables from four locations in the Canadian Arctic Archipelago (CAA), obtained from both observations and models.  The observations presented are very interesting, giv- ing potentially a unique view of the evolution of ice thickness in the CAA, as well as some insight into the climate and climate change in the CAA. The data, results, and conclusions are not ground breaking or spectacular, but can serve as a solid addition to our knowledge of the Arctic. The authors also use the observational data to evaluate model results. This is a useful exercise, but still needs some work. The paper is mostly clear and concise. The authors' approach in working with the data and models is also good in general.  There are a number of points that need to be addressed before I can recommend publication, but I am confident that the authors can address those in a satisfactory manner. Given that I expect the resulting paper to become a good, relatively low key, but solid addition to our understanding of the CAA climate.

My main concern regarding this paper is with the conclusions the authors draw at the end.  These are too often poorly supported by the data, or even not at all.  In some cases the discussion is lacking so that it is not clear what conclusions to draw.  The most  confusing  aspect  is  the  role of  snow  and  temperature,  where  it  is  not  always clear when we should be thinking about de-trended correlations or correlation between trends.   It is also not clear how you calculate the correlation between maximum ice thickness and temperature and it's not immediately clear how this should be done (do you want to use a mean temperature over the growing season for instance?)

In particular, in the first paragraph of the conclusions the authors state that "[e]ven though warming is seen at all sites,  changes in ice thickness is also attributable to variability in snow depth,  which plays a dominant role in controlling the inter-annual mean and variability of ice thickness". Here the authors appear to be mixing the long- term trend in temperature (the warming) with the inter-annual variability,  which they show is highly correlated with snow thickness (the de-trended correlation is correctly used for this). It seems to me that the inter-annual variability in maximum ice thickness is controlled by variability in snow thickness and that the long-term trend is caused by a long term trend in temperature. But I'll come back to this point below.

The authors then go on to say "[w]ithin the CAA, increases in snow depth are con- tributing to decreased trends in maximum ice thickness at Eureka and Alert but this far appear to be exerting less of an impact on maximum ice thickness at Resolute and Cambridge Bay". I don't understand how the authors reach this conclusion. The trend in snow thickness is only significant at

Cambridge Bay so we can safely assume that only there can a trend in snow thickness contribute to a trend in ice thickness.

Turning to the relationship between ice thickness and temperature, in paragraph four of the conclusions the authors state: "the significant correlations at Resolute, Eureka and Alert suggest that the higher sensitivity to changes in snow depth could easily mask the warming signal on both fast and offshore ice". I don't really feel this has been shown in the paper. I very much expect this statement to be true, but it needs to be better argued.

The authors then go on to say "[t]he dependency between ice thickness trends and warming trends is only weakly present at Cambridge Bay (r= 0.4)", but in the text it is clearly stated that the given r-value is related to the de-trended correlation and has as such nothing to do with the thickness and warming trends.

My conclusions, after reading the paper (and not your conclusions) are that
• Snow thickness is the main driver of inter-annual variations in maximum ice thick- ness (high de trended correlation at two out of three sites, and reasonably high correlation at the other two).
• Inter-annual variations in air temperature are only weakly correlated with maximum ice thickness (max r= 0.4).
• A trend in air temperature is the main driver of the trend in maximum ice thick- ness. This is because there is a significant warming trend at all sites, but only significant snow thickness trend at one.

In short the inter-annual variations in maximum ice thickness are caused by variations in snow, while the long term trend is caused by a long term trend in temperature. The fact that there is no ice thickness trend at Resolute is probably related to the timing of snow fall, but this is only mentioned once in the paper and not explored in any depth. The authors conclude that change in snow thickness, not temperature is the driver of both inter-annual and long-term changes, but I can only agree with the former part of that statement, not the latter.

Now, I don't have access to all the data and haven't spent much time on the analysis, so maybe I'm completely wrong and the authors right. But if this is the case then the authors need to make a more convincing point in their presentation of the results and conclusions.

**Howell et al.,**
**1. We agreed that perhaps the conclusions a little too speculative given the results of our analysis. We have removed the speculate statements in the first paragraph and changed the text. We essentially agree with the reviewers first 2-conclusions points. However, we do not agree that the warming from air temperature is the main driver of maximum ice thickness decline because of the low correlations with temperature. The first paragraph of the conclusion is now as follows:**
*Over the 50+ year in situ observational record, negative trends in maximum (end-of-winter) ice thickness are found at all four sites with statistically significant trends present at Cambridge Bay, Eureka and Alert. Negative trends in the day of maximum ice thickness are also present at all sites and statistically significant at Resolute, Eureka and Alert. Together, these trends suggest thinning of landfast ice in the CAA, where little evidence was found in the*

*shorter record analyzed in an earlier study (BC92). The inter-annual variability of air temperature is only weakly correlated to maximum ice thickness (i.e. maximum correlation is ~0.4). Snow thickness plays the dominant role in controlling maximum ice thickness variability given the high correlations at Resolute and Eureka and reasonably high correlations at Alert and Cambridge Bay.*

**2. With respect to snow cover masking the warming signal, we do feel this a reasonable conclusion that can be drawn since it is clear from this analysis as well as previous literature that snow cover is the dominate factor in controlling landfast ice thickness. We have changed the final paragraph of the paper to be more conservative but still reflect this:**

*While the impact of the snow cover on ice thickness is well known, the significant correlations at Resolute, Eureka and Alert suggest that the higher sensitivity to changes in snow depth could potentially mask the warming signal on both fast and offshore ice. Thus, even in this limited data set, we can see the dominant role played by snow depth in determining the interannual variability of the maximum landfast ice thickness. This again highlights that the primary factor is the amount and timing of snow accumulation, not air temperature. However, it is worth noting that few of the current generation models show coherent relationships between ice thickness, snow depth and temperature over the longer term record.*

**3. We use temperature of the same months shown in Figure 7. This is now clearer in the text and corrected based on a specific comment by the reviewer.**

In addition I'm also concerned over sections 4.2, and 4.3. First of all, the ORA-IP models are all run over a different, and much shorter time period than is covered by the observations. I therefore doubt that it is appropriate to use those to calculate trends that should be compared to the observed trends. In addition the first paragraph of that section left me very confused. You propose comparing figures 2 and 12, but the former shows thickness seasonal cycle, while the latter shows thickness trends. You talk about a general pattern, but this is too vague and qualitative for my taste. Also, I'm not sure there is a north-south pattern in the observed trends, we can at least not see this from table 2. In short, it seems to me you should focus on CMIP5 for the trend analysis because the ORA-IP periods are too short. It could be OK to use the longest runs, i.e. CGLORS, ORAP5.0, and PIOMAS (that runs from 1979, right?), but be careful.

**Howell et al.**
**We have modified section 4.2 to better reflect our results and we are confident that it now gives better the reader appreciation of which trends are significant and which one are not. We have however kept the ORA-IP models and their discussion because we considered that we had sufficiently careful to ensure that significant results were robust through our use of the False Discovery Rate for trend patterns. This means that even though some patterns show few stippling, these stippling can (and probably should) be considered robust and reflective of a model characteristic. For example, in the correlation discussion, most ORA-IP show large regions of positive correlations and two models show that these positive correlations are likely significant, at least for some grid points. While this is far from the**

**end of the story, we consider this behavior to be an important departure from the in situ observations that it should be clearly highlighted in the current study.**

**First paragraph of 4.2:** *The spatial distribution of maximum sea ice thickness trends from ORA-IP and CMIP5 is illustrated in Figures 12. The CMIP5 model-mean exhibit a fairly uniform trend pattern, consistent with the different in situ observations (Figure 4) but with overestimated negative thickness trends. Although, for individual models this pattern is far from uniform, the general pattern and magnitude of thickness trends tend to be roughly in accordance with temperature trends (not shown). A similar behavior is observed in the ORA-IP models, with the notable exception of CGLORS, where positive thickness trends are found almost everywhere (Figure 12a). This is robust and it appears that the model is not completely equilibrated in the CAA and exhibit large month-to-month adjustments. Model ORAP5.0 also is not completely equilibrated in the region for years 1979-1984. During those years, it exhibits large inter annual changes in thickness. For this reason, we are only considering years 1985-2013 for this model.*

2 Specific comments
l. 25
There is hardly a need for a reference to what land-fast ice is, but if you want one I would recommend the WMO definition. On the other hand, I would like to see a reference for the statement "this ice typically extends to the 20–30 m isobath". It is right, but should be backed up by some references (I think Hajo Eicken recently published a nice overview and you already have Mahoney et al. in your reference list).

**Howell et al.**
**Agreed. First two sentences changed as follows:**
*The World Meteorological Organization (WMO, 1970) defines landfast sea ice as "sea ice which remains fast along the coast, where it is attached to the shore, to an ice wall, to an ice front, or over shoals, or between grounded icebergs." In the Arctic, this ice typically extends to the 20-30 m isobaths [Mahoney et al., 2007; Mahoney et al., 2014].*

l. 101
Here you say that the CMIP5 runs extend from 1980 to 2099, but later you say that you use the period 1955–2014. But doesn't the historical run go back to 1850?

**Howell et al.**
**There was a typo in the models description and we are grateful that the reviewer pointed it out. To answer the reviewer's question, yes, these simulations go back to 1850 but we have chosen to use the same period as the observations.**

l. 117
I would add PIOMAS to table 2 for completeness.

**Howell et al.**
**Done. Table follows the revised Figures at end of this document.**

l. 162

You can do better than suggesting a shortened growth season. In the conclusions you cite Howell et al. who show a later onset of melt and if you combine this with your results you are conclusively demonstrating a shortened growth season.

**Howell et al.**
**Changed to reflect the reviewer's suggestion as follows:**
*Typically, ice thickness reaches its maximum in late May with trends toward earlier dates of maximum ice thickness present at all sites (significant at Resolute, Eureka and Alert; Table 3). The significant trends are between -2.0±0.1 days decade$^{-1}$ at Eureka to -6.2±1.5 days decade$^{-1}$ at Resolute. At Resolute, the date of maximum ice thickness is now on average more than a month earlier than the early 1960's although this is not reflected in the trend in ice thickness. Freeze onset at these sites is also increasing at ~3-6 days decade$^{-1}$ [Howell et al., 2009] and demonstrates a shortened growth season at Resolute, Eureka and Alert. Together, the trends of ice thickness and their recorded dates suggest a systematic thinning of landfast ice at Cambridge Bay, Eureka and Alert.*

l. 171
You should not tell us which trends there are, only to then tell us that they are all insignificant except one. In fact you should only discuss the significant trends and if you want to mention the others then don't spend to much time on them.

**Howell et al.**
**Agreed and removed the wording with respect to insignificant trends. Changed to:**
*For snow depth, the only trend that is statistically significant at the 95% confidence is Cambridge Bay at -0.8±0.4 cm decade-1 (Table 3).*

l. 179
Here you say that there is statistical correspondence between the ice and snow thickness trends at Eureka and Alert, but the snow thickness trends there are not significant. This does not make sense to me. Also, it is not clear what you mean by "statistical correspondence".

**Howell et al.**
**Changed as follows:**
*Figure 6 provides evidence from extreme years of the role of deeper snow inhibiting ice growth compared to thinner snow, but the positive trends in snow thickness are not significant at Resolute, Eureka and Alert. This may in part be due to the single pointwise snow depth and ice thickness measurements made at each point in time, which fail to capture spatial heterogeneity in the snow depth/ice thickness relationship.*

l. 186
I couldn't understand the sentence starting on this line. Please rewrite.

**Howell et al.**
**Changed as follows:**
*The detrended correlation between temperature (winter, spring, summer and autumn) and maximum ice thickness is weak at all sites. For example, the strongest detrended correlation*

*between maximum ice thickness and temperature (winter and spring) is found at Cambridge*
*Bay during the winter and spring but is only ~0.4.*

l. 207
For completeness you should note Parry Channel in figure 1. But I'm not sure about this
condition, since ocean and ice dynamics presumably play a small role here. If you want a more
quantitative measure than "a reasonable representation of the CAA" then I would suggest putting
a limit on the resolution in the CAA.

**Howell et al.**
**Added location of Parry Channel in Figure 1. The Parry Channel plays a large role as the**
**majority of ice flow within the CAA takes place within the Parry Channel in terms of**
**exchange with Arctic Ocean and Baffin Bay.**

[Figure]

l. 215
Picking single points to analyse is always a bit questionable. You should make sure, and state
that the point selection does not affect your results, as long as the points selected follow your
selection criterion. I suppose that there were more than one candidates for each model and
location. Also, some of the model points are quite far away from the observation station. I noted
especially points representing Alert and Cambridge Bay which are so far away from the stations
that I would expect them to belong to a different atmospheric and oceanographic regime (see
figure 12f).

**Howell et al.**
**We have now added a few sentences at the end of the paragraph that better justify why we**
**used sample points. We are now stating that we have checked the seasonal cycles at the**
**sample points using a modified selection criterion and that we found no quantitative**
**differences. Also, we think that it is important to show that some models have perennial ice**
**in regions where observations show none (Figure 12f, e.g. the samples for Alert, in black,**
**should probably sit on top of the red x for all models), in the hope that it would lead to an**
**improved depiction of CAA landfast ice in future sea ice models. While it is true that this**
**comparison to sample points puts models to high standard, we think that such a difficult**
**benchmark might be desirable as sea ice models are increasingly resolving the narrow**
**channels of the CAA.**

**Added sentences:**
*This is a consequence of using samples as some models either do not resolve some of the*
*channels in the CAA or have too perennial packed ice cover (e.g. CESM1-CAM5), then the*

*sample points are further from the observational site than would be desired. We chose to use sample points in our comparison to observations instead of using regional averages for two main reasons. The first reason is that using regional averages would have lumped together different ice dynamics regimes that should not necessarily be expected to compare well to point observations on landfast ice. The second reason is that we are of the opinion that the resolution in many of these models is sufficiently high to warrant such a direct comparison and provides a better benchmark than regional averages for landfast ice modelling in the CAA.*

l. 221
Here you say that the seasonal cycle is calculated over 1955–2014, but this does not fit your earlier statement about the CMIP5 models or the information in Table 2.

**Howell et al.**
**As mentioned earlier, this was fixed in the model description. Thank you for highlighting this discrepancy.**

l. 235
It is not clear to me why this statement should be true. Please elaborate.

**Howell et al.**
**We have rewritten this statement. It now reads:** *This likely reflects the fact that the grid cell thickness in sea ice models with thickness classes a represents the average thickness over these classes. In August the thinner ice classes might have melted but thicker ice classes can still be found, resulting in a substantial average ice thickness over the grid cell.*

l. 264
Which models are the "high resolution models"?

**Howell et al.**
*This paragraph was rewritten and these words were removed. Originally, the high resolution models were all the models that are resolving the Parry Channel, as explained in 4.1*

l. 276
In table 2 the trend is 4.3±1, not 6.2±2.4. There is also an inconsistency in signs in this paragraph.

**Howell et al.**
**They were computed over the PIOMAS record. In this analysis we used 1979-2014. This is now clear in the text.**

l. 289
If the trends are not significant then we can't learn much from them.

**Howell et al**

**We have rephrased to emphasize that while these correlations are only significant at a few locations in a few of the ORA-IP models, it so counterintuitive and potentially problematic that we feel it is important that a red flag be raised.**

l. 295
Again, trends that are not significant don't warrant much of a discussion.

**Howell et al**
**We have reformulated these sentences to remind the reader that trends in snow depth are not significant. However, we disagree with the reviewer that one cannot discuss non-significant trends. It is our opinion that in the results presented here, strong, significant snow depth trends in models are in stark contrast with the weak non-significant trends observed over the last six decades at the in situ locations.**

l. 301
Don't talk about the insignificant trends as if they matter.

**Howell et al.**
**Changed as follows:** *Over the 50+ year in situ observational record, statistically significant negative trends in maximum (end-of-winter) ice thickness are present at Cambridge Bay, Eureka and Alert. Significant negative trends in the day of maximum ice thickness are also present at Resolute, Eureka and Alert*

l. 324
This paragraph doesn't belong in your conclusions since it contains nothing from the current work.

**Howell et al.**
**Removed.**

3    Technical corrections
l. 20 Replace "two magnitudes" with "two orders of magnitude".

**Howell et al.**
**Done.**

l. 131 Replace "2.27 m that is likely ..." with "2.27 m, which is likely ...'

**Howell et al.**
**Done.**

l. 169 Here it sounds as if it's the linkages that are summarised in table 3, but this is not the case.

**Howell et al.**
**Removed text mentioning that.**

l. 176 The r should be italicised and the equations would look much nicer if the were within dollar signs (assuming you're using latex).  i.e. r=−0.66, instead of r=-0.66.

**Howell et al.**
**We are not using latex. We assume this will be changed according to the copyediting protocol used by The Cryosphere.**

l. 193
Replace "than the earlier period" with "from the earlier period".

**Howell et al.**
**Done.**

l. 213
Replace "good portion of year but not all year" with "good portion of the year, but not all year"

**Howell et al.**
**Done.**

Figure 7: I would think that having separate panes for each season would make more sense. There should be much less difference between the temperature at different locations than there is between the different seasons. That way we can more easily see the trend. As it is all the lines look very flat.

**Howell et al.**
**Indeed the trend is easier to see but the graph is very messy. We initially tried this but the overlap is even worse and it is more difficult see the separation for each site. Although the line is flat this is the best way to show the temperature data.**

Figure 8: I would like to see the observations on these figures as well. They may become a little crowded, I'll admit, but it's important to be able to see, at a glance how the models depart from observations.

Figures 9 & 10: The observations should be shown here as well.

**Howell et al.**
**Figures modified to include observations.**

[Figure]

Figure 8. CMIP5 median sea ice thickness seasonal cycle (1955-2014) at stations (grey). Observations from 2 (black). Median of ORA-IP models CGLORS, ORAP5.0, GLORYS2V3 (blue), ECCO-v4 (green) and UR025.4 (red). Whiskers indicate the 5th and 95th percentiles.

[Figure]

Figure 9. Same as Figure 8 for snow depth and only for CMIP5 models (grey) and observations (black).

Table 2. Summary of reanalysis models characteristics

| Model Name | CGLORS | ECCO-v4 | GLORYS2V3 | ORAP5.0 | UR025.4 | PIOMASS |
|---|---|---|---|---|---|---|
| **Institute** | CMCC | JPL-NASA-MIT-AER | Mercator Océan | ECMWF | University of Reading | APL/PSC |
| **Resolution** | ORCA0.25° | ~40km in the Arctic | ORCA0.25° | ORCA0.25° | ORCA0.25° | ~22km in the Arctic |
| **Ocean Model** | NEMO 3.2.1 | MITgcm | NEMO 3.1 | NEMO3.4 | NEMO 3.2 | POP |
| **Sea ice Model** | LIM2 | MITgcm | LIM2 (with EVP rheology) | LIM2 | LIM2 | TED |
| **Time period considered** | 1982-2012 | 1991-2011 | 1993-2013 | 1985-2013 | 1993-2010 | 1958-2015 |
| **Atmospheric forcing** | ERA-Interim | ERA-Interim | ERA-Interim | ERA-Interim | ERA-Interim | NCEP/NCAR |
| **Sea ice product assimilated** | NSIDC NASA-Team Daily | NSIDC Bootstrap Monthly | IFREMER/CERSAT | NOAA / OSTIA combination | EUMETSAT OSI-SAF | NSIDC near-real time Daily |

---

## Author Comment (AC2) · 2 Jun 2016

Reviewer 1
This manuscript describes an analysis of ice thickness and snow depth from a combination of local observations and from CMIP models. Overall, I believe this article was well written and would be of interest to readers of the Cryosphere journal. I do have one minor suggestion for the authors. Instead of trying to pick model points closest to the observed sites, perhaps a better approach is to average over the whole Canadian Arctic Archipelago. As the authors point out, the CMIP models do not have the resolution to properly represent the channels in this area. So, a larger area average might be more meaningful. This could be compared to the average of the four observed locations. Also, perhaps some thoughts on how the relationships between land fast ice and snow depth could be better represented in the models would be helpful.

**Howell et al.**
**We have pondered the reviewer's suggestion of using area averages instead of sample points but decided against it for two main reasons that are now explicitly spelled out in the manuscript based on suggestions from Reviewer 2. One of the reasons is that taking a regional average would have lumped together very different ice regimes, something we really wanted to avoid. Another reason is that we wanted to provide a benchmark for current and upcoming high resolution sea ice models. We are of the opinion that the higher resolution models should be held to a high standard and should be able to roughly describe the observed seasonal cycle at these observational sites. Regarding discussion on improving snow depth in models, we feel it would be too speculative on our part and outside the conservative nature of our conclusions.**

---

## Author Response (AR2)

Dear colleagues,

thank you for the revision of the manuscript. I still have some minor issues before I can accept the manuscript for final publication.

1) I tried to get the data from the website http://www.ec.gc.ca/glaces-ice/ to reproduce the plots. It seems the data are distributed over two tables and it is not clear for me how to merge both into a consistent time series. Moreover, only for Alert there are data for the period 2003 until present. Thus, it was not possible for me to reproduce your results. Could you please add the data used for the analysis to a supplement? It would be good if you could also include the supporting air temperature from reanalysis. Please note the data source in Fig. 7.

**Howell et al.**
**The stations we used (Cambridge Bay, Resolute, Eureka and Alert) have indeed remained unchanged over the long term record and merging them is very easy because the date of the measurement is given. In order to add value to our paper we have included a merged Excel spreadsheet for both ice and snow thickness. We have also included a spreadsheet for the temperature data which is observed by Environment Canada and not reanalysis.**

**For this observed data we have added in the data description that the data is available in "supplementary material" We have changed the caption in Figure 7 to reflect the data source as well as in the Data Description.**

**Data Description Changes:**
*"The other source of observed data used in this study was Environment Canada's monthly mean air temperature records at Alert, Eureka, Resolute, and Cambridge Bay (see supplementary material) for which a complete description is provided by Vincent et al. [2012]."*

*"Values of maximum or end-of-winter ice thickness and corresponding snow depth during the ice growth season were extracted from the weekly ice and snow thickness data at the selected sites (see supplementary material)."*

**New Figure 7 Caption:**
*Figure 7. Time series observed mean air temperature by Environment Canada during winter (DFJ), spring, (MAM), summer (JJA) and autumn (SON) at the Cambridge Bay, Resolute, Eureka and Alert.*

2) Section "Models" includes the methods of statistical analysis mixed with the climate/ocean models. Please distinguish between the trend analysis and the description of climate/ocean models and change the structure. The "methods" are now under the section "Data description". You may add another section "Methods" to describe the statistical test for significance in more detail. I have not fully understood your procedure for the test of significance and would like to learn more about the noise model. The reference to the two papers is not enough to reproduce your results.

**Howell et al.**
**We modified the Data Description Removing lines 109-116 replacing them with the following:**
*"We obtain the multi-model mean of trends and their statistical significance at each grid point by creating the distribution of trends through a Monte-Carlo simulation. We use a t-distribution for the interannual variability and build a noise model to account for internal variability as in Swart et al. [2014] and Laliberté et al. [2016]. We obtain the multi-model mean of Pearson correlations and their statistical significance by first performing a Fisher transform and then applying the same method as for the trends. The inverse Fisher transform is applied after obtaining the multi-model mean and its significance. See the appendix for a complete description of the method."*

We then added an Appendix describing our statistical approach as follows:
*"The Monte-Carlo simulation used to combine trends and Pearson correlations is applied at each grid point independently. Models that have a land mask at a grid point are discarded before starting the procedure.*

*A noise model is created to ensure that internal variability is comparable for models with different ensemble sizes, following Swart et al. [2014] and Laliberté et al. [2016]. To generate the noise model, we discard models that have fewer than two realizations. From the remaining models, we pick one and then one of its realizations. We then record to the noise model the difference of this realization's trend from the mean trend of the model's realizations, multiplied by $(n/(n-1))^{1/2}$, with n being the number of realizations, to account for the fact that some models have such a small number of realizations that it cannot completely account for the internal variability. We repeat this procedure 1000 times and compute the variance $\sigma_n$ of the noise model.*

*We then pick a model from which we select 1000 realizations, allowing repetitions. For each one of these realizations, we select a random value from its trend t-distribution. If the inter-realization trend variance $\sigma_m$ is smaller than the variance of the noise model $\sigma_n$, we then draw a random value from the noise model, multiply it by $(1-\sigma_m/\sigma_n)^{1/2}$ and add it to the random value from the trend t-distribution.*

*We repeat this procedure with the remaining models. We then average the 1000 values across models, creating a distribution for the multi-model mean trend with 1000 values. The mean of this distribution gives our multi-model mean and its two-sided p-value is given by twice its survival function or cumulative distribution function at 0, whichever is smallest.*

*The Pearson correlations are analyzed in the same way except that a Fisher transform (obtained by the hyperbolic arctangent of the correlation) is applied first and random values are drawn from a normal distribution (instead of the t-distribution) with variance $1/(T-3)$, with T the number of years used for the correlation. The multi-model mean Pearson correlation is then given by the inverse Fisher transform (obtained by the hyperbolic tangent of the mean) of the distribution mean."*

3) COV is usually used for covariance, at least I do so. Please use the more common CV for coefficient of variation.
**Howell et al.**
**Changed.**

4) Table 2) PIOMASS -> PIOMAS
**Howell et al.**
**Changed.**

**Additional Note:**
**We have included a new version of Figure 14. The original text referred to this version of the Figure but we included an older version by mistake in the original submission.**